# Examining Changes in Consumer Spatial Structure and Sustainable Development Issues in Beijing before and after the Outbreak of COVID-19

**Changsheng Shi, Bin Meng \*, Yuting Yuan, Zhiyuan Ou and Xiaohang Li**

College of Applied Arts and Sciences, Beijing Union University, Beijing 100191, China;
changshengshi@foxmail.com (C.S.); 20221070510116@buu.edu.cn (Y.Y.); 20221070510101@buu.edu.cn (Z.O.);
xiaohangli2019@163.com (X.L.)
\* Correspondence: mengbin@buu.edu.cn

**Abstract:** Urban consumption spatial structure has a direct impact on the sustainable development and quality of life of urban residents. This study investigates the impact of COVID-19 on Beijing's urban consumption spatial structure within the Sixth Ring Road. Utilizing POI (Point of Interest) data and the Kernel Density method, the spatial distribution of commercial centers is analyzed. Consumption data from China UnionPay for 2019 and 2020, along with the Weighted Voronoi diagram method, are employed to assess changes in the radiation range of commercial centers. The findings indicate that: (1) owing to the pandemic's repercussions, commercial centers at different levels and locations have different changes. (2) There is an overarching decline in UnionPay consumer spending across diverse streets in Beijing. (3) Following the epidemic, large-scale consumption hubs have maintained their dominance, ensuring the overall stability of the city's consumption spatial pattern. In conclusion, the changes in commercial centers and the significant decrease in consumer spending underscore the dynamic interplay between urban consumption and external shocks like the pandemic. These insights are crucial for urban planning strategies aiming to enhance both resilience and sustainability in the face of unforeseen challenges. The spatial restructuring of traditional commercial centers requires nuanced urban planning. Recognizing the resilience and expansion of smaller centers suggests the importance of fostering localized economic activities. Policymakers could incentivize their development to promote community engagement and economic sustainability.

**Keywords:** COVID-19; consumption space structure; UnionPay consumption data; weighted Voronoi diagram

## 1. Introduction

Urban consumption spatial structure constitutes a pivotal facet of sustainable urban development [1]. Serving as a primary arena for social and economic activities, urban consumption spaces have long been a focal point of geographical research. A judicious layout of commercial establishments within a city can optimize its spatial structure, cater more effectively to the daily needs of residents, and foster economic development [2].

Since the outbreak of COVID-19 in December 2019, an unprecedented health crisis has precipitated substantial shifts in residents' consumption behavior and psychology, bringing the nation's economic activities to a standstill [3]. Nevertheless, through the implementation of a series of epidemic policies and guidelines, China has achieved a resounding and decisive victory in epidemic prevention and control, leading to a recovery in the economic environment [4]. The urban consumption spatial structure of major Chinese cities has undergone profound changes in the wake of COVID-19. Beijing, as the capital and a hub for commercial activities, serves as a noteworthy case study to scrutinize the alterations in urban consumption spatial structure before and after the outbreak. This

exploration provides a theoretical foundation for reconfiguring Beijing's consumption spatial pattern in the post-epidemic period.

Researching the spatial pattern of urban consumption has long been a concerned for the geographical community. From the perspective of the structural system of consumption space, as early as 1994, Yang used the central place theory to explore the formation mechanism, spatial structure and reasonable prediction of the retail and service industry in the Beijing metropolitan area [5]. In 2001, Wu et al. constructed an index system and used geographic information system (GIS) technology and hierarchical clustering analysis to classify and describe the structural types of a Beijing Business Center in detail [6]. Zhang et al. compared Beijing with some foreign capital cities, determined the development stage of the Beijing Central Business District, analyzed the spatial structure problems in the process of urban development, and predicted the development trend [7]. In addition, a variety of geographic analysis methods and data are also used by many geographers to study the characteristics of urban consumption space from multiple perspectives and predict the development trend of consumption space. Lin et al. used the POI data of retail outlets, the concept of "function" and the systematic clustering method to establish the hierarchical system of the Beijing Business Center [8]. Hou et al. used the cumulative occupancy data on Sina Weibo to analyze the characteristics of Shanghai's commercial spatial structure and its location-related factors and put forward suggestions to strengthen the connections between suburban new towns and Shanghai's central urban rail transit so as to promote the development of Shanghai's suburban commercial centers [9]. Wang et al. used Moran's I, high/low class, clustering and outlier analysis, principal component analysis, geographically weighted regression and other spatial analysis methods and research methods to study the evolution characteristics and spatial distribution patterns of an urban online celebrity culture consumption space. By constructing the principal component index system based on three categories, this paper analyzes the influence mechanism of the spatial distribution of urban online celebrity culture consumption and deeply understands the characteristics and development trend of an urban online celebrity culture consumption space [10].

In the process of urban development, the changes in urban consumption spatial structures directly affect the daily consumption behavior modes of people. In the period of the COVID-19 outbreak, the change of this consumption behavior pattern is particularly obvious. Li et al. explored the impact of the COVID-19 epidemic on residents' consumption behavior and its formation mechanism by taking residents' families as a unit [11]. Eger l et al. studied the changes in consumer behavior patterns during the second wave of the new coronal pneumonia epidemic in the Czech Republic and the impact of new coronal pneumonia on consumer purchase behavior based on questionnaire data [12]. Zhang et al. proposed a two-way high-resolution framework based on financial transaction records and population flow data to assess the epidemiological and economic results of different blockade policies during epidemic prevention [13]. Lu et al. analyzed the new characteristics and changes in the consumption field before and after the epidemic from the perspective of price index and per capita consumption expenditure and combined these characteristics with the reality of Suzhou, providing policy suggestions concerning upgrading the consumption structure in Suzhou under normalized epidemic prevention and control [14].

Summarizing previous research, it is evident that there is a paucity of studies examining the changes in consumption spatial patterns in large Chinese cities before and after the onset of the unique phase of the COVID-19 epidemic. Previous studies predominantly focused on static and single-mode consumption spatial structures or selectively extracted specific consumption activities for spatial structure analyses. The research data and scope were often limited, primarily concentrating on particular business districts. In contrast, our study encompasses a comprehensive array of consumer activities. It is essential to recognize that the spatial structure of consumption and people's consumption behavior are interconnected. Therefore, leveraging big data, this study delves into the analysis of residents'

consumption behavior before and after the outbreak of the epidemic, aiming to encapsulate the alterations in consumption spatial structure characteristics across commercial centers in Beijing. The findings of this study bear significant implications for understanding the transformation of consumption spatial structures and fostering sustainable development in Beijing during the post-pandemic era.

## 2. Data and Method

### 2.1. Study Area

As the capital and national central city of China, Beijing has a long history of commercial development and has always represented the course of China's economic development, bearing profound marks of the times [15]. Since the establishment of Dadu in the Yuan Dynasty, commercial activities have been carried out in Beijing [5]. For thousands of years, Beijing has always played an important commercial role, witnessing the prosperity and development of China's economy. In this city, commercial activities are thriving, with all kinds of markets, trading places and merchants gathering, bringing together people from different regions and cultural backgrounds, creating rich and colorful business exchanges and cultural integration. Its commercial centers and outlet layouts thus became the prototype of Beijing's early urban consumption spatial structure, whose influence continues to this day [5]. In recent years, with the advancement of urbanization, the area of Beijing's urban districts has continued to expand, and suburban areas have been fully developed, while the area within the Sixth Ring Road has become the main object of research on consumption spatial structures [16,17].

According to reports from the Beijing Municipal Bureau of Statistics, by the end of 2020, Beijing's permanent resident population reached 21.893 million, of which the urban population was 19.166 million. The permanent population within Beijing's Sixth Ring Road reached 17.077 million, accounting for 78% of the total permanent population of the city [18]. Therefore, this study takes the area within Beijing's Sixth Ring Road as the research scope, covering nearly 80% of the city's permanent residents. Spatially, the research scope includes the two core urban districts of Dongcheng and Xicheng, as well as parts of the four suburban districts of Chaoyang, Haidian, Fengtai and Shijingshan adjacent to the urban area, and it also covers parts of the five outer suburban districts of Changping, Tongzhou, Daxing, Fangshan and Shunyi (Figure 1).

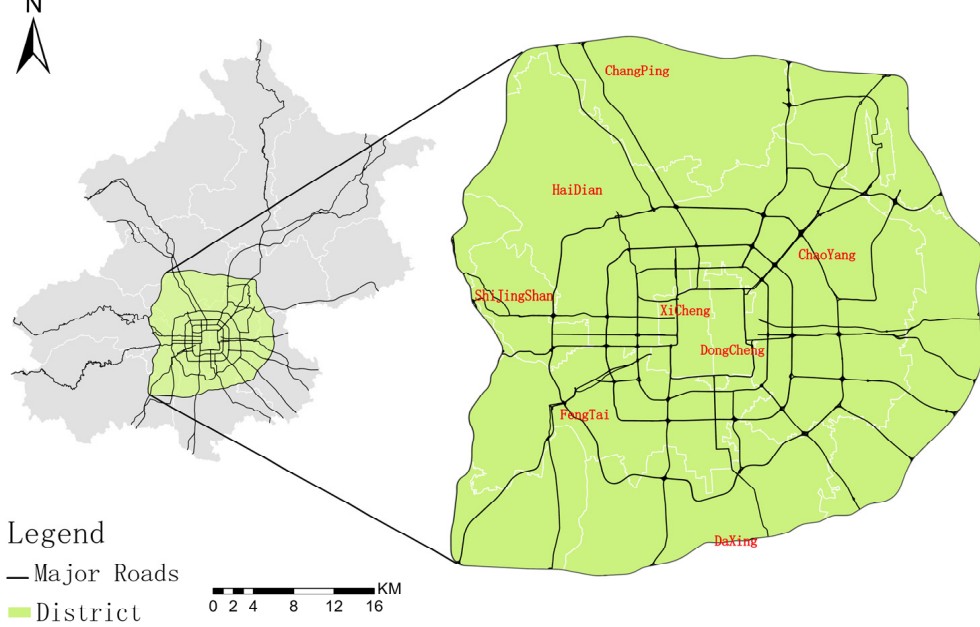

**Figure 1.** Study area.

### 2.2. Data

2.2.1. UnionPay Data

China UnionPay (China UnionPay Co., Ltd. Beijing Branch, Beijing, China) is a state-owned financial institution established in 2002 and headquartered in Beijing. As a payment and clearing organization, UnionPay plays an important role in China's payment field. By the end of 2021, the cumulative number of accepting merchants exceeded 490 million, and the acceptance coverage continued to expand. At the same time, UnionPay's services extensively cover multiple areas closely related to people's daily lives, including transportation, retail, catering, tourism, education, etc. [19]. The UnionPay card payment method is widely used in major cities across China and has become one of the mainstream payment methods for people's travel and shopping [20]. UnionPay card transaction data not only include shopping information such as payment amounts, product types, and transaction date, but also massive geographic location information (Table 1), which makes it easy to study urban consumption patterns, consumption attenuation ratios, and changes in the radiation range of commercial hotspots.

**Table 1.** Data dictionary of UnionPay card swiping data.

| Field | Type | Length | Description |
|---|---|---|---|
| id | Int4 | 32 | Commercial Network ID. |
| Lon | Float8 | 53 | Longitude of the Commercial Network. |
| Lat | Float8 | 53 | Latitude of the Commercial Network. |
| Total consumption amount | Float8 | 53 | Total Daily Expenditure at the Commercial Network. |
| Entertainment_amount | Float8 | 53 | Daily Expenditure on Entertainment at the Commercial Network. |
| Daily necessities_amount | Float8 | 53 | Daily Expenditure on Daily Necessities at the Commercial Network. |
| Catering Category_Amount | Float8 | 53 | Daily Expenditure on Dining at the Commercial Network. |
| General service category_Amount | Float8 | 53 | Daily Expenditure on General Services at the Commercial Network. |
| Entertainment_number of entries | Float8 | 53 | Number of Entertainment Expenditure Transactions at the Commercial Network. |
| Daily necessities_number of transactions | Float8 | 53 | Number of Daily Necessities Expenditure Transactions at the Commercial Network. |
| Catering Category_Number of Transactions | Float8 | 53 | Number of Dining Expenditure Transactions at the Commercial Network. |
| General services_Number of transactions | Float8 | 53 | Number of General Services Expenditure Transactions at the Commercial Network. |

Using the location query function of ArcGIS Pro 2.5, a total of 1123 UnionPay card swiping spots were screened within Beijing's Sixth Ring Road in May 2019, and 1106 in May 2020 (Figure 2).

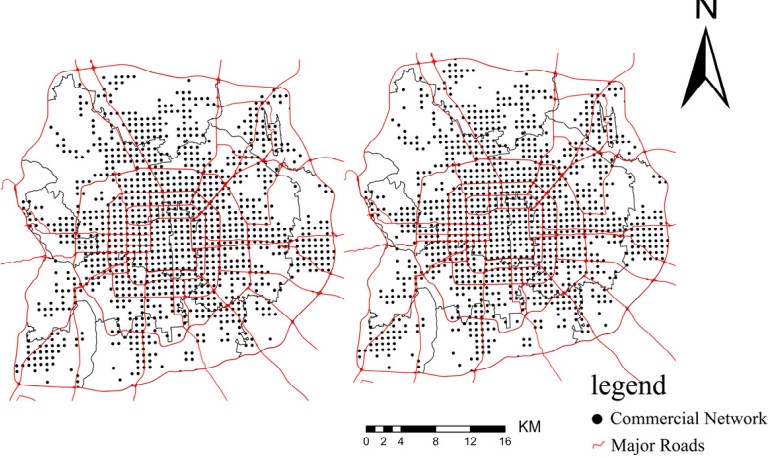

**Figure 2.** UnionPay card swiping records within the Sixth Ring Road (19 years on the left, 20 years on the right).

### 2.2.2. Commercial POI Data of Beijing

In addition to UnionPay data, this study employs the commercial POI data of Beijing in 2019 and 2020 from Amap, from which 393,731 and 117,284 commercial outlets within the Sixth Ring Road were extracted for 2019 and 2020. Meanwhile, relevant geographical data, including Beijing's district and subdistrict boundaries, main traffic routes, and other related geographic information data, are combined to conduct the research.

### *2.3. Method*

### 2.3.1. Spatial Hotspot Analysis

This study determines Beijing's commercial centers at all levels based on commercial POI data using the kernel density estimation (KDE) method and the density-field-based hotspot detector (DF-HD).

Kernel density estimation is a widely used method for spatial data analysis, which is used to estimate the density distribution of point data within their surrounding neighborhood. The calculation formula is as follows (1). Details about KDE can be found in References [21,22].

$$f_n(x) = \frac{1}{nh}\sum_{i=1}^{n} k\left(\frac{x - x_i}{h}\right) \tag{1}$$

Although continuous digital field models generated based on density surfaces such as kernel density maps can express the spatial density distribution characteristics of POI from an overall perspective, they can hardly accurately obtain hotspot areas and peak values, which affects subsequent research.

A more effective method is to conduct hotspot analysis and pattern analysis through quantitative expression and description of hotspot areas or peak values. Such a quantification method can more precisely identify and represent the hotspot areas of POI, helping us gain an in-depth understanding and explanation of the characteristics and trends in the study area [23]. This study adopts the density-field-based hotspot detector model, in which grid cell extreme value zones are extracted from the density field grid map as hotspot centers using methods such as window analysis and map algebra difference operation, and the extremes are used as the peak values of the hotspots. See the Section 3.3 of commercial center selection and determination for details.

### 2.3.2. Weighted Voronoi Diagram

In a Voronoi diagram, the space is partitioned into different regions based on a set of points in two or three dimensional space. Each region consists of the point closest to the region's interior (also called a "seed point") and all adjacent points to that point (Figure 3a) [24]. A weighted Voronoi diagram refers to assigning corresponding weights to each point on the basis of the Voronoi diagram. These weights can determine the centroid position of the point within the polygon. In a weighted Voronoi diagram, each point has a certain weight, and the larger the weight, the greater the influence of that point, which in turn affects the shape and size of the Voronoi diagram where that point is located (Figure 3b) [25]. The size of a weighted Voronoi diagram region can reflect the importance and radiation range of the points within that region.

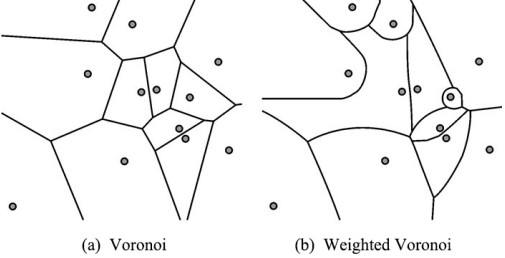

      (a) Voronoi        (b) Weighted Voronoi

**Figure 3.** Voronoi and Weighted Voronoi (Adapted with permission from Ref. [25]. 2000, Chakroun, H.; Benie, G.B.; O'Neill, N.T.; Desilets, J.) (**a**) Traditional Voronoi; (**b**) Weighted Voronoi.

For any point set $P = \{P_1, P_2, \ldots, P_n\}$ ($n \geq 3$) in a plane, let $\lambda_i$ ($i \geq 1$) be the attribute factor corresponding to each point. Then, the region $V_n(P_i, \lambda_i)$ determined by the point $P_i$ and factor $\lambda_i$ ($i \geq 1$) divides the plane into n parts [25]. The plane partitioning method determined by $V_n(P_i, \lambda_i)$ ($i \geq 1$) is called the weighted Voronoi diagram of the points. The formula is as follows (2).

$$V_n(P_i, \lambda_i) = \bigcap_{i \neq j} \left\{ P \mid \frac{d(P, P_i)}{\lambda_i} \leq \frac{d(P, P_i)}{\lambda_j} \right\} (i, j = 1, \cdots, n) \tag{2}$$

where $\lambda_i$ represents the weight of point $P_i$, and for each point in each partitioned region, the ratio of its distance to the center of that region to its distance to the center of the adjacent region is less than the ratio of the weights of the two centers [26]. The Voronoi diagram can be viewed as a special case of the weighted Voronoi diagram when the weights are equal [27].

This study constructs the radiation ranges of commercial hotspots using a weighted Voronoi diagram and analyzes the spatial patterns of consumption hotspots based on these ranges.

## 3. Results and Analysis

In this section, we use UnionPay data and POI to analyze how the consumption spatial structure of Beijing changed after the COVID-19 outbreak. The analysis encompasses the spatial pattern of consumption hotspots, the consumption attenuation ratio, and the radiation range of the consumption hotspots.

### 3.1. Spatial Pattern of Consumption Hotspots

This study utilizes UnionPay data from May 2019 and May 2020 to explore changes in the spatial pattern of consumption hotspots through spatial overlay and spatial hotspot analysis. This method allowed for the precise spatial identification of consumption hotspots and cold spots, contributing to a better understanding of the pandemic's impact on consumption.

Based on the kernel density results of UnionPay data (shown in Figure 4), from a macro perspective, although COVID-19 has had a certain impact on the consumer market, large-scale consumer centers in Beijing are still dominated by traditional commercial centers such as Xidan, Sanlitun, Wangfujing, Financial Street, CBD, Taiyang Palace, Huilongguan, Tiantongyuan, Shangdi, and Yizhuang. In the past two years, the overall consumption spatial pattern has not changed significantly.

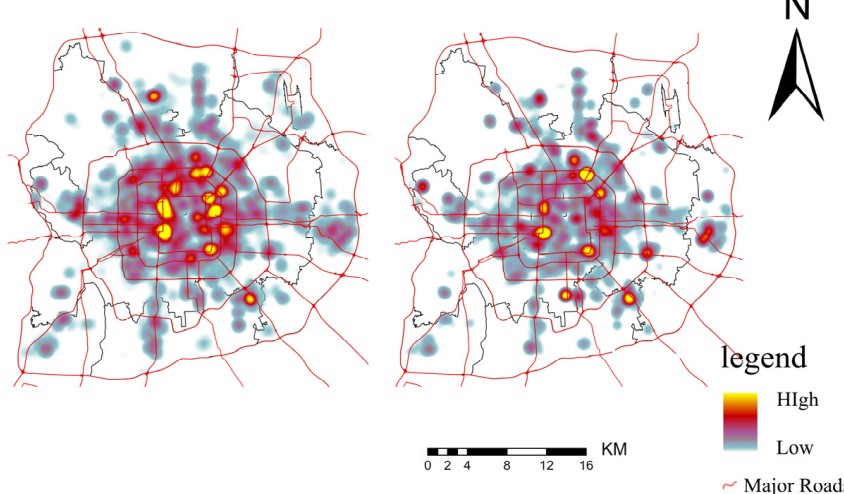

**Figure 4.** Consumption density of UnionPay in May (**left**) 2019 and May 2020 (**right**).

On a micro level, we can see that different levels of commercial centers have undergone some changes over the past two years. For city-level commercial centers, the kernel density values of areas such as Sanlitun, CBD, Chaowai and Dongzhimen have decreased compared to 2019, and their influence has also decreased. For regional-level commercial centers, the kernel density values of some commercial centers located in the east and west urban areas within the Fourth Ring Road have also decreased compared to 2019, such as Jishuitan, Gongzhufen, Wukesong, Shuangjing, Wanliu and Yaao. Meanwhile, regional commercial centers far from the city center, such as Wangjing, Chaocheng, Yizhuang, Tongzhou Business District, Shangdi, Huilongguan, and Tiantongyuan, saw an increase in kernel density values, indicating that due to COVID-19, Beijing residents are more willing to go to the nearest regional-level commercial center for consumption, and are less likely to choose regional-level commercial centers near the city center but far away from their residences.

Overall, although Beijing's large-scale consumption centers are still dominated by traditional commercial centers such as Xidan, Sanlitun, Wangfujing, Financial Street, CBD, Sun Palace, Huilongguan, Tiantongyuan, Shangdi, and Yizhuang, the spatial pattern of consumption has undergone some subtle changes in the past two years.

In addition, in terms of functional positioning, commercial centers that mainly provide leisure and entertainment services, such as the Ya'ao Business Circle and the Beijing Zoo, have seen their kernel density values decrease compared to 2019, while commercial centers that are closer to the community and mainly provide life-type services, such as small- and medium-sized supermarkets around the community, have seen their kernel density values increase. It can be seen that due to the impact of COVID-19, residents have intentionally reduced their travel consumption for leisure and entertainment purposes, and their daily consumption is mainly based on the acquisition of daily necessities.

The outbreak of COVID-19 had a limited impact on the overall spatial pattern of Beijing's consumer hotspots. However, at a local level, the epidemic still has an impact on the level of commercial centers, the distance from the city center, and the functional positioning of commercial centers to a certain extent. It further shapes the city's spatial structure, which also reflects the city's adaptability and changing trends in the face of emergencies.

### 3.2. Changes in Consumption Pattern

COVID-19 has a serious impact on people's production and life, and the reduction in people's travel behavior has directly led to a decline in the city's consumption level. In order to explore this change, this paper uses UnionPay consumption data to calculate the decay rate of residents' consumption and compares it with the government's monthly statistical bulletin. In addition, we spatially analyze the consumption attenuation ratios of each street within the Sixth Ring Road of Beijing as a means of exploring the changes in the urban spatial pattern of consumption after the outbreak of COVID-19.

#### 3.2.1. Attenuation Ratio of UnionPay Card Expenditure

By aggregating and calculating the UnionPay card transaction expenditures in Beijing for the year for 2019 and 2020, it was determined that the total UnionPay card expenditure in Beijing for May 2019 was approximately CNY 113,812,506,660, while in 2020 it was approximately CNY 59,330,616,710. The reduction in expenditure from 2019 to 2020 was approximately CNY 54,481,889,950, with an attenuation ratio of approximately 47.87%. These data illustrate that during the pandemic, residents significantly reduced their outdoor activities, leading to a substantial decrease in the total UnionPay card expenditure.

According to the "Beijing Statistical Yearbook 2020" [28] (Table 2) released by the Beijing Municipal Bureau of Statistics, it can be observed that in May 2020, compared to May 2019, the total retail sales of consumer goods experienced a year-on-year decrease of 9.3%. Specifically, when categorized by industry, the accommodation and catering sector saw a year-on-year decline of 70.7%, while the consumption form-wise categorization revealed a 38% decrease in dining revenue and a 6.8% decrease in commodity retail. This indicates that due to the

pandemic, urban residents in Beijing significantly reduced their outdoor activities, aligning with the results reflected by the UnionPay card transaction data.

**Table 2.** Total retail sales of consumer goods in Beijing in May 2020.

| Item | 2020/05 | Year-on-Year Ratio (%) |
|---|---|---|
| **Retail Sales of Consumer Goods** | **10,441,834** | **−9.3** |
| Including: Online retail sales of wholesale and retail businesses, accommodation and catering businesses | 2,928,183 | 24.1 |
| **By Commodity Use** | | |
| Food Products | 2,054,977 | −12.8 |
| Clothing | 542,453 | −30.9 |
| Consumer Goods | 7,481,242 | −3.5 |
| Durable Goods | 363,162 | −41.5 |
| **By Industry** | | |
| Wholesale | 1,392,971 | −4.5 |
| Retail | 8,471,491 | −7.2 |
| Accommodation | 37,584 | −70.7 |
| Catering | 539,788 | −32.7 |
| **By Region** | | |
| Urban | 9,921,962 | −9.5 |
| Rural | 519,872 | −5.5 |
| **By Consumption Form** | | |
| Dining Revenue | 577,372 | −38 |
| Retail Sales of Goods | 9,864,462 | −6.8 |

### 3.2.2. Attenuation Ratio of Street-Level Consumer Expenditure

Due to the impact of COVID-19, the urban consumption pattern of Beijing has also changed significantly. Although the overall trend is downward from the perspective of total consumption, at the spatial level, different spatial locations show their own unique attenuation changes. In order to explore this change more accurately, this study takes the street as a scale and aggregates and superimposes 19 and 20 years of UnionPay consumption data into each street and analyzes the consumption attenuation of each street within the Sixth Ring Road by calculating the value of the attenuation ratio. The smaller the value of the attenuation ratio, the more the street's spending amount declined over the 20-year period and has a stronger trend of attenuation, which is calculated as follows:

$$Attenuation\ Ratio = \frac{20 - year\ expenditure - 19 - year\ expenditure}{20 - year\ expenditure} \quad (3)$$

As shown in Figure 5, the consumption attenuation ratios of streets within the Sixth Ring Road show a negative value, and compared with 2019, more than half of the streets have an attenuation of more than 50%, while only a small number of streets have a positive value, indicating that due to the impact of the epidemic, the out-of-town consumption willingness of urban residents has dropped significantly, and the amount of consumption has decreased accordingly. Specifically, the consumption attenuation of streets within the Sixth Ring Road shows three patterns: First, the closer the street is to the city center, the lower its consumption attenuation ratio is, and the consumption attenuation ratio of streets within the Fourth Ring Road is generally lower than that outside the Fourth Ring Road; second, the closer the street is to the city's main arterial road, the lower its consumption attenuation ratio is, and the attenuation ratio of streets that are farther away from the main arterial road is comparatively larger; third, the streets that are farther away from the city center were relatively less affected by COVID-19 and had relatively larger consumption attenuation ratios, with some streets having positive values.

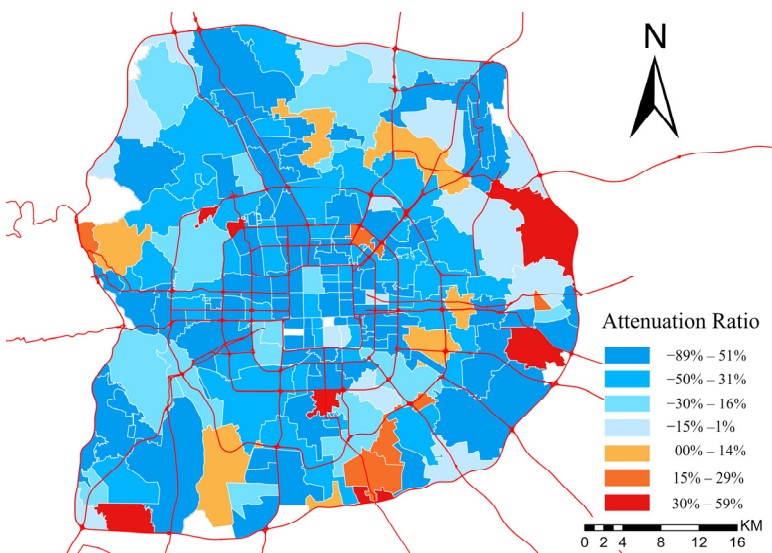

**Figure 5.** Consumption attenuation ratio of each street before and after the outbreak of COVID-19.

### *3.3. Changes in the Radiation Range of the Consumption Hotspots*

In order to explore the structural changes in Beijing's consumption space after COVID-19, this study adopts the weighted Voronoi method to draw Voronoi diagrams and uses the UnionPay expenditure as weights, thereby illustrating the spatial radiation range of each commercial center.

This study adopts the following method in the selection and determination of commercial centers: firstly, a kernel density map is drawn using the POI data of commercial outlets in Beijing, and then, with the help of the density field hotspot detector, different levels of commercial centers are determined according to the pixel values of the kernel density map. The specific process is shown in the following Figure 6:

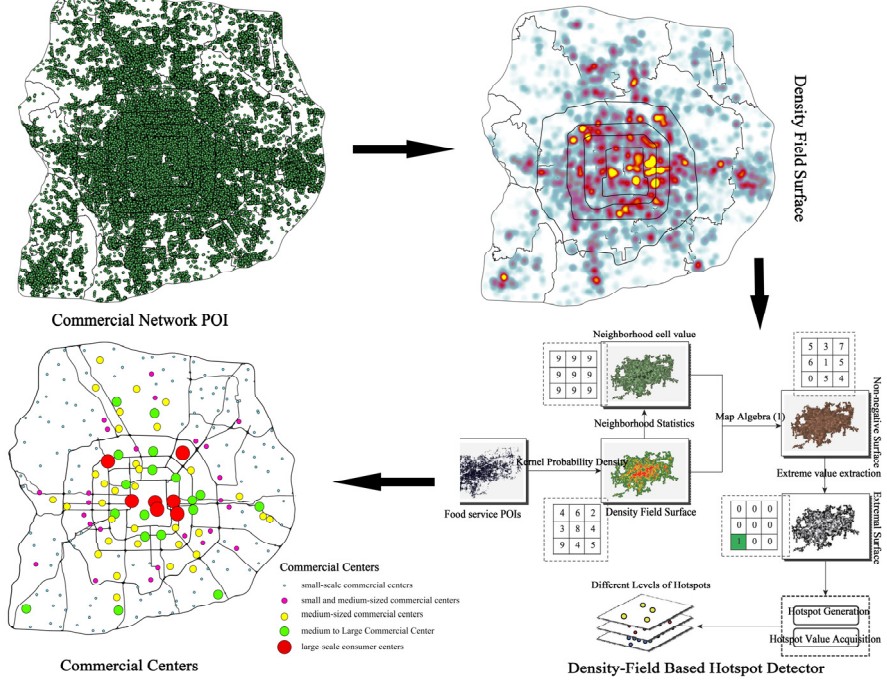

**Figure 6.** Extract commercial center flow chart.

As shown in the figures (Figure 7), in comparison to 2019, there was a decrease in the number of commercial centers in 2020. Particularly, some small-scale commercial centers were adversely affected by the COVID-19 pandemic, with a continuous decline in their customer numbers, ultimately resulting in the disappearance of these commercial centers. However, medium and large-scale commercial centers, leveraging their inherent size and robustness, remained stable under the impact of the COVID-19 pandemic, showing minimal changes in their numbers.

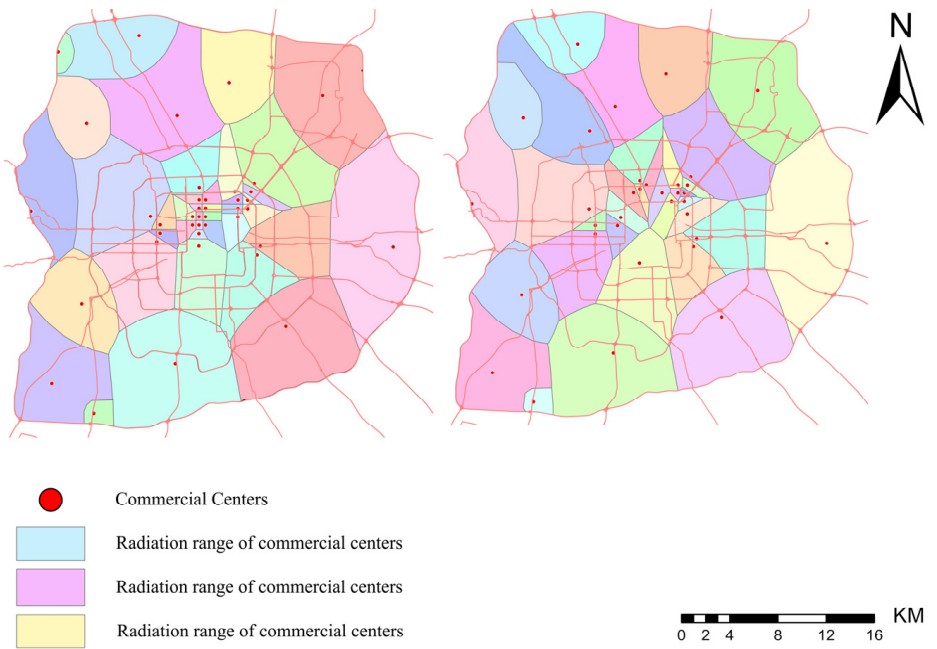

**Figure 7.** Radiation range of commercial centers in 2019 (**left**) and 2020 (**right**).

Regarding the radiation range of commercial centers, different levels of commercial centers exhibit distinct and unique trends. Influenced by the pandemic, the radiation range of traditional large-scale commercial centers has been continuously shrinking, while the small and medium-sized commercial centers far away from the city center and closer to the community have shown an expanding trend, which is mainly due to the fact that urban residents have drastically reduced the number of outings, and the purpose of going out has also changed from the consumption of leisure and entertainment to the purchase of necessities, and the frequency of consumption of luxury goods and large-scale valuables has even dropped significantly. So, residents prefer small and medium-sized commercial centers near their communities to traditional large commercial centers with high traffic volume when they go out shopping. From a spatial perspective, the radiation range of commercial centers in the southern region within the sixth ring road exhibits no significant change compared to the pre-pandemic period. In the northern region, the radiation range has expanded. In the central region, particularly around the east and west city areas, the radiation range has undergone noticeable changes, with an overall expansion of the radiation range for each commercial center.

In addition, for a more intuitive observation of changes in the radiation range of consumption hotspots, this study conducted a profile analysis of consumption in selected hotspot areas (Figure 8). Specifically, we chose three profile lines along the Airport Expressway, the west–east axis of Xidan, and the north–south axis of Xidan, on which we plotted consumption Sectional view for the years 2019 and 2020. The Airport Expressway extends from the Third Ring Road at Sanyuan Bridge, connecting the Third Ring Road, Fourth Ring Road, and Fifth Ring Road before intersecting with the Airport South Line and arriving at Capital International Airport Terminals 1 and 2.

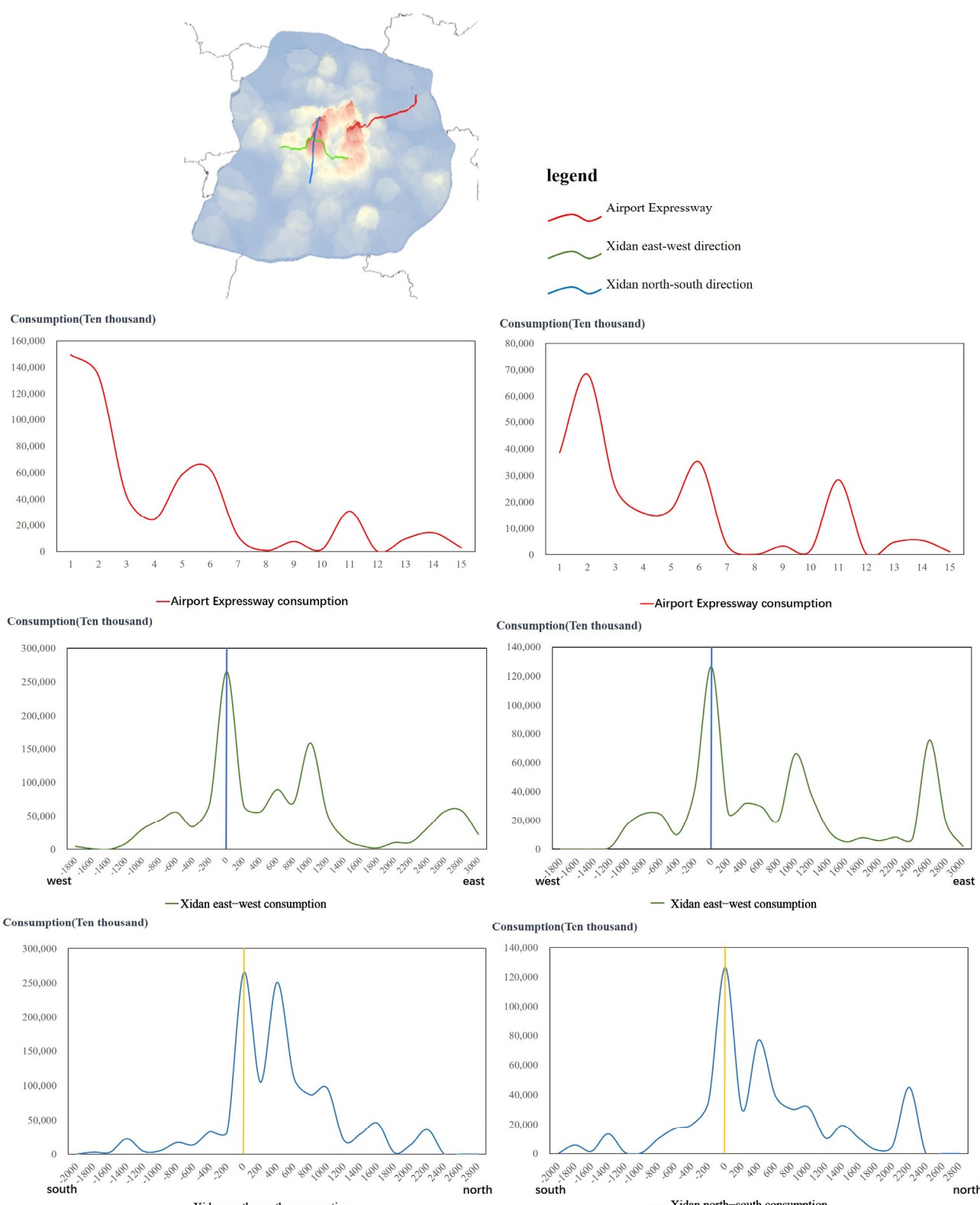

**Figure 8.** Sectional view of consumption hotspots (left is 2019, right is 2020).

From a sectional view, it is evident that apart from an overall expenditure decrease of over 50%, the undulating trend along the Airport Expressway profile exhibited minimal change over the two years. Notably, the proportion of expenditures near Sanyuan Bridge has decreased considerably, while the proportion of expenditures around Wuyuan Bridge and Wenqu Bridge has increased. This indicates that, under the influence of the pandemic,

the radiation strength of the Sanyuan Bridge commercial center along the Airport Expressway has weakened, while the radiation strength of smaller commercial centers relatively farther from the city center has risen.

Looking at the Xidan west–east profile, there are no significant changes in expenditure profiles within a range extending westward or up to 2400 m eastward. However, beyond 2400 m to the east, the proportion of expenditures has significantly increased. This suggests that the Xidan commercial center, impacted by the pandemic, has experienced a reduction in both its eastward radiation range and radiation intensity, while its westward radiation range and intensity have remained largely unchanged. Regarding the north–south profile of Xidan, the radiation range of the Xidan commercial center toward the south remains essentially unchanged. Within the range of 200 to 400 m to the north, there is a slight increase in radiation intensity. However, at a greater distance of 2000 m to the north, there is a decrease in radiation intensity. This indicates that the northward radiation range of the Xidan commercial center has decreased compared to 2019.

To sum up, due to COVID-19, the spatial structure of consumption in Beijing has undergone more obvious changes. In addition to the changes in the number of commercial centers, commercial centers in different spatial locations and different grades have different change situations, and at the same time, through the drawing of the consumption sectional view of specific regions, it has been found that the radiation strength of each commercial center to different directions also has different change situations.

## 4. Discussion

### 4.1. Key Findings

This study takes the range within the sixth ring road of Beijing as the study area, and with the help of POI data of commercial outlets according to the spatial distribution and degree of aggregation of commercial facilities, uses the kernel density analysis method combined with the hotspot detection model of density field to extract the commercial centers in Beijing before and after the outbreak of epidemic, and at the same time combines with the UnionPay consumption data of 2019 and 2020, and applies Weighted Voronoi diagram to analyze the post-epidemic spatial changes in the radiation range of commercial centers. By comparing the changes in the number and location of commercial centers, as well as the differences in their radiation ranges, we explore and analyze the changes in the spatial structure of consumption. Our key findings are as follows:

Firstly, the spatial structure of traditional commercial centers has been affected by the epidemic to varying degrees from the perspectives of their functions, hierarchy, and proximity to the city center: the impact of the epidemic has led to the disappearance of a number of commercial centers and a trend toward a reduction in the radiation range of traditional large commercial centers. In contrast, small and medium-sized commercial centers that are far away from the city center but closer to the community have expanded their radiation range after the outbreak, showing certain survival and development advantages.

Secondly, in terms of the amount of consumption, the consumption level of Beijing residents dropped significantly after the outbreak, and the overall consumption showed a downward trend. At the spatial level, streets close to the city center and major arteries were more significantly affected by the decline in consumption amount. On the other hand, streets located near the Sixth Ring Road and relatively far away from the city center were less affected by the epidemic, with only a slight decline in consumption and even an upward trend in some streets.

Finally, on the whole, after the outbreak, the main consumption hotspots of the residents are still concentrated in the traditional large commercial centers, and the overall spatial pattern of consumption in Beijing has remained basically stable, which indicates that the evolution of urban commercial centers is a long and complex process, and it requires a long and detailed research process to explore the evolution of the spatial pattern of consumption as well.

*4.2. Practical Implications*

The primary significance of this study lies in two aspects. Academically, we introduced an innovative research perspective by exploring the changes in the spatial structure of urban consumption from three dimensions: spatial pattern of consumption hotspots, consumption attenuation ratio, and the radiation range of the consumption hotspots, revealing the impact of the epidemic from three perspectives. Furthermore, we use multi-source big data and spatial analysis methods, which are scientific and efficient. In future studies, this methodology can be easily extended to other cities for consumption space structure and sustainable development studies, thereby providing methodological support for the broader field of urban and social geography and sustainability.

In practice, the findings in this study have several policy implications. First, our study reveals that the spatial restructuring of traditional commercial centers post-epidemic necessitates a nuanced approach to urban planning and development. Recognizing the resilience and expansion of small and medium-sized commercial centers closer to communities suggests the importance of fostering localized economic activities. Policymakers may consider incentivizing the development of such centers to promote community engagement and economic sustainability. Secondly, the observed decline in the consumption levels of Beijing residents post-epidemic indicates a need for targeted economic stimulus measures. Policymakers could explore initiatives such as consumer incentives, tax breaks, or other financial stimuli to revitalize consumption, particularly in areas closer to the city center and major arteries that experienced a more significant downturn. Conversely, areas less affected by the epidemic could benefit from strategic investments to maintain their positive consumption trends. Furthermore, the study underscores the importance of a flexible and adaptive urban development strategy. The stability of consumption hotspots in traditional large commercial centers suggests a certain level of enduring consumer behavior. Policymakers should consider measures to enhance the resilience of these centers, potentially through infrastructural improvements, business support programs, or cultural initiatives that attract residents back to these commercial hubs. In conclusion, the policy implications drawn from our findings emphasize the need for a dynamic and context-specific approach to urban planning and economic revitalization. As cities adapt to the long-term impacts of the COVID-19 pandemic, policymakers should consider a mix of strategies tailored to the evolving spatial patterns of commercial activities and consumption behaviors, ensuring a resilient and sustainable urban development trajectory.

## 5. Conclusions

Urban consumption spatial structure is a complex concept that greatly influences the livability and sustainability of cities. This study reveals the changes in the spatial structure of consumption in Beijing after the outbreak of COVID-19 from multiple perspectives of consumption. We first analyzed the spatial changes of consumption hotspots through POI and UnionPay data, then calculated the consumption decay ratio, and finally analyzed the changes in the radial range of consumption hotspots using weighted Tyson polygons. Our results show that COVID-19 has had a large impact on the spatial structure of urban consumption, and there is significant spatial heterogeneity in consumption patterns. Overall, our study provides valuable insights into the structure of urban systems. Inevitably, this study has some limitations: First, this study solely relies on UnionPay data to indicate the consumption situation, lacking other data for verification. In terms of the methodology, the use of only the Weighted Voronoi diagram to represent the radiation range of commercial centers, with consideration given only to consumer spending as a weight, may have limitations. In subsequent research, we plan to enhance the accuracy by incorporating various methods to map the radiation range. Additionally, other mobile payment data should be combined to provide a more comprehensive understanding of the consumption patterns. Secondly, although the changes in the spatial structure of consumption are analyzed from three aspects, they are all limited to urban areas and ignore the differences between urban and rural areas. In future research, it will be necessary to introduce the urban–rural

dichotomy and strengthen the exploration of urban–rural differences to further explore the differences in the quality of life between urban and rural areas through changes in the spatial structure of consumption [29]. Finally, we also need to continue to pay attention to the impact of urban planning and public policy and explore how to make full use of the trend of changes in the spatial structure of consumption in the process of policy formulation and implementation so as to promote the sustainable development of cities.

**Author Contributions:** Conceptualization, C.S. and Y.Y.; methodology, B.M.; software, C.S.; validation, Z.O., Y.Y. and X.L.; formal analysis, C.S.; investigation, C.S.; resources, B.M.; data curation, C.S., Z.O. and X.L., Writing—original draft preparation, C.S.; writing—review and editing, Y.Y.; visualization, C.S. All authors have read and agreed to the published version of the manuscript.

**Funding:** This research was supported by the Academic Research Projects of Beijing Union University (Grant No. ZKZD202305), National Natural Science Foundation of China (Grant Nos. 41671165).

**Institutional Review Board Statement:** Not applicable.

**Informed Consent Statement:** Informed consent was obtained from all subjects involved in the study.

**Data Availability Statement:** The data presented in this study are available on request from the corresponding author. The data are not publicly available due to privacy.

**Conflicts of Interest:** The authors declare no conflict of interest.

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
