# Peer review of "Examining Changes in Consumer Spatial Structure and Sustainable Development Issues in Beijing before and after the Outbreak of COVID-19"

_sustainability, doi:10.3390/su152316451_

Round 1

Reviewer 1 Report

Comments and Suggestions for Authors

Nice written paper overall. Except minor changes, one big issue is there is no data after the pandemic in the raw data and the data analysis, which make the paper logically not making sense.

Reviewer 2 Report

Comments and Suggestions for Authors

The manuscript possesses several notable strengths. Firstly, the abstract provides a concise and comprehensive overview of the study, effectively highlighting the importance of comprehending urban consumption spatial structure for the sake of sustainable development. It adeptly establishes a connection between the research and the influence of COVID-19 on Beijing's traditional consumption spatial structure while introducing the employment of advanced geographic data analysis methods in the research methodology.

Secondly, the introduction is praiseworthy for laying a solid foundation. It accomplishes this by not only reviewing previous research but also providing a detailed contextual background of the study area.

Furthermore, the methods employed in the research, including kernel density estimation, density-field-based hotspot detection, and Voronoi diagrams, are well elucidated and visually supported, which enhances clarity and understanding for readers.

In addition, the study's utilization of real-world data from May 2019 and 2020 to scrutinize alterations in consumption patterns is highly relevant, especially in light of the pandemic's impact.

Lastly, the Conclusion and Discussion section serves as a succinct summation of the findings, effectively showcasing the research's comprehensive and rigorous approach, with clear implications for informing sustainable development policies.

However, there are certain shortcomings in the manuscript that should be noted as the followings:

Abstract 

The abstract is somewhat lengthy and could be more concise. It includes a significant amount of detail about the research methods and findings, which could be reserved for the main body of the paper.

While it mentions the impact of the pandemic on consumer concerns and spending trends, it does not offer specific insights or implications based on these findings. A more explicit discussion of the implications for urban planning and policymaking would be valuable.

The abstract does not include a statement regarding the limitations of the study or any mention of potential areas for future research, which could provide context and direction for readers.

Some phrases could be clearer and more specific, such as "sustainable reshaping of Beijing's consumption space structure." It would be helpful to provide more detail on what this entails.

Introduction

The introduction lacks a clear research objective or research questions. While it states the study's focus on changes in Beijing's consumption spatial structure, it doesn't specify what the study aims to achieve or investigate.

Some sentences are overly long and complex, making the text difficult to read and understand. For instance, sentences 39 to 56 are quite lengthy and could benefit from more concise and straightforward language.

There is a lack of transition sentences between different sections of the introduction, which can make the flow of the text disjointed. The transition from discussing previous research to the study's focus could be smoother.

The citation of previous research is extensive, which may be overwhelming for readers, and it could be more focused on works directly related to the current study's scope.

The introduction does not explicitly state the methodology or approach that will be used in the research, leaving the reader without a clear understanding of how the study will be conducted.

Data and Method

While the section is informative, it is quite lengthy and could benefit from more concise and straightforward language. It may be challenging for some readers to grasp all the details due to the extensive technical explanations.

The section lacks a clear outline or structure, making it somewhat difficult to follow. It could benefit from subheadings to break down the information into smaller, more manageable sections.

The use of formulas and mathematical notations, while relevant to the methodology, may be challenging for non-experts or those without a strong background in spatial analysis.

The section could include more information on potential limitations and challenges associated with the chosen methods, as this would provide a more balanced view of the research approach.

Results and Analysis

The section is quite lengthy, and while it provides comprehensive information, it could benefit from more concise language to avoid overwhelming the reader.

The text contains some typographical errors and inconsistent formatting (e.g., "Attenuation Datio" instead of "Attenuation Ratio").

While there is a focus on presenting quantitative data, the interpretation of qualitative aspects, such as residents' motivations and reasons for reduced outdoor activities, could provide a more holistic understanding.

The section could benefit from a summary or a brief overview at the beginning to provide readers with a roadmap of the key findings before delving into the details.

Some of the specific terms, such as "radiation influence," may benefit from clearer definitions or explanations to ensure that readers fully understand the concepts used in the analysis.

Conclusion and Discussion 

The conclusion could be more explicit in summarizing the main implications of the findings and their significance for policymakers, urban planners, and other stakeholders.

While the study identifies changes in the spatial structure of consumption, it falls short of providing concrete recommendations or suggestions for addressing the challenges or opportunities revealed by these changes. A more specific discussion of potential strategies or actions for the future would be beneficial.

The section mentions the need for further research and exploration but does not delve into any potential future research directions or questions that could build upon this study.

Some parts of the discussion are quite broad, such as the statement that "evolution of urban commercial centers is a long and complex process," which, while true, may not provide actionable insights for readers.

Comments on the Quality of English Language

Spelling checks are required.

Reviewer 3 Report

Comments and Suggestions for Authors

Dear Authors

congratulation to your paper.

I was in  Beijing in 2018 and 2023. It is very a complicated and interesting topic.

Please correct structure of your article.

4. Discussion

5. Conclusion

I like your tabs, results and analysis.

if it is possible  implement connection between urban-rural dichotomy,  quality of life / and / with your article: 

Petrovič, F.; Maturkanič, P. Urban-Rural Dichotomy of Quality of Life. Sustainability 2022, 14, 8658. https://doi.org/10.3390/su14148658,

correct your structure

Round 2

Reviewer 3 Report

Comments and Suggestions for Authors

paper is accepted.